# A Personal Breast Cancer Risk Stratification Model Using Common Variants and Environmental Risk Factors in Japanese Females

**DOI:** 10.3390/cancers13153796

**Published:** 2021-07-28

**Authors:** Isao Oze, Hidemi Ito, Yumiko Kasugai, Taiki Yamaji, Yuko Kijima, Tomotaka Ugai, Yoshio Kasuga, Tomoyo K. Ouellette, Yukari Taniyama, Yuriko N. Koyanagi, Issei Imoto, Shoichiro Tsugane, Chihaya Koriyama, Motoki Iwasaki, Keitaro Matsuo

**Affiliations:** 1Division of Cancer Epidemiology and Prevention, Aichi Cancer Center Research Institute, Nagoya 464-8681, Japan; ymaeda@aichi-cc.jp (Y.K.); tugai@aichi-cc.jp (T.U.); kmatsuo@aichi-cc.jp (K.M.); 2Division of Cancer Information and Control, Aichi Cancer Center Research Institute, Nagoya 464-8681, Japan; hidemi@aichi-cc.jp (H.I.); y.taniyama@aichi-cc.jp (Y.T.); ykoyanagi@aichi-cc.jp (Y.N.K.); 3Division of Descriptive Cancer Epidemiology, Nagoya University Graduate School of Medicine, Nagoya 466-8550, Japan; 4Division of Cancer Epidemiology, Nagoya University Graduate School of Medicine, Nagoya 466-8550, Japan; 5Division of Epidemiology, Center for Public Health Sciences, National Cancer Center, Tokyo 104-0045, Japan; tyamaji@ncc.go.jp (T.Y.); moiwasak@ncc.go.jp (M.I.); 6Department of Breast Surgery, School of Medicine Fujita Health University, Aichi 470-1192, Japan; ykijima@fujita-hu.ac.jp; 7Department of Pathology, Brigham and Women’s Hospital, and Harvard Medical School, Boston, MA 02115, USA; 8Department of Epidemiology, Harvard T.H. Chan School of Public Health, Boston, MA 02115, USA; 9Department of Surgery, Nagano Matsushiro General Hospital, Nagano 381-1231, Japan; ykasuga@hosp.nagano-matsushiro.or.jp; 10Kaneko Clinic, Kagoshima 890-0055, Japan; tmyoknko@gmail.com; 11Director, Aichi Cancer Center Research Institute, Nagoya 464-8681, Japan; iimoto@aichi-cc.jp; 12National Institute of Health and Nutrition, National Institutes of Biomedical Innovation, Health and Nutrition, Tokyo 162-8636, Japan; stsugane@nibiohn.go.jp; 13Division of Cohort Research, Center for Public Health Sciences, National Cancer Center, Tokyo 104-0045, Japan; 14Department of Epidemiology and Preventive Medicine, Kagoshima University Graduate School of Medical and Dental Sciences, Kagoshima 890-8544, Japan; fiy@m.kufm.kagoshima-u.ac.jp

**Keywords:** breast cancer, genetic risk model, polygenic risk model, environmental risk model, personalized prevention

## Abstract

**Simple Summary:**

Breast cancer remains the most common cancer in females, warranting the development of new approaches to prevention. One such approach is personalized prevention using genetic risk models. Here, we developed a risk model using both genetic and environmental risk factors. Results showed that a genetic risk score defined by the number of risk alleles for 14 breast cancer risk SNPs clearly stratified breast cancer risk. Moreover, the combination of this genetic risk score model with an environmental risk model which included established environmental risk factors showed significantly better C-statistics than the environmental risk model alone. This genetic risk score model in combination with the environmental model may be suitable for stratifying individual breast cancer risk, and may form the basis for a new personalized approach to breast cancer prevention.

**Abstract:**

Personalized approaches to prevention based on genetic risk models have been anticipated, and many models for the prediction of individual breast cancer risk have been developed. However, few studies have evaluated personalized risk using both genetic and environmental factors. We developed a risk model using genetic and environmental risk factors using 1319 breast cancer cases and 2094 controls from three case–control studies in Japan. Risk groups were defined based on the number of risk alleles for 14 breast cancer susceptibility loci, namely low (0–10 alleles), moderate (11–16) and high (17+). Environmental risk factors were collected using a self-administered questionnaire and implemented with harmonization. Odds ratio (OR) and C-statistics, calculated using a logistic regression model, were used to evaluate breast cancer susceptibility and model performance. Respective breast cancer ORs in the moderate- and high-risk groups were 1.69 (95% confidence interval, 1.39–2.04) and 3.27 (2.46–4.34) compared with the low-risk group. The C-statistic for the environmental model of 0.616 (0.596–0.636) was significantly improved by combination with the genetic model, to 0.659 (0.640–0.678). This combined genetic and environmental risk model may be suitable for the stratification of individuals by breast cancer risk. New approaches to breast cancer prevention using the model are warranted.

## 1. Introduction

Breast cancer is the most common cancer in females, with an estimated global incidence in 2018 of 2,088,849 [1]. Breast cancer is also a leading cause of death worldwide, causing 15.1 million disability-adjusted life years (DALY) in 2016 [2]. Furthermore, incidence is estimated to increase to 3,059,829 cases in 2040 [1]. In Japan also, breast cancer incidence has increased rapidly for the last 30 years and is now the most common cancer in women [3]. This increase is associated with changes in the prevalence of established risk factors in Japanese women, which might be broadly categorized as behavioral and social westernization.

Conventional strategies for breast cancer prevention include control of risk factors and early detection through mammography screening, targeting women in the general population. Given the increased burden of breast cancer, however, the development of new prevention strategies is essential. As a revolutionary approach, personalized prevention or precision prevention has recently been proposed [4,5,6]. A detailed elucidation of individual breast cancer risk might allow personalized intervention in women stratified by risk factors. Several breast cancer risk prediction models have been developed to evaluate individual risk based on lifestyle factors, reproductive factors, family history and clinical factors [7,8,9,10,11]. These are now in clinical use as tools for individual cancer prevention. For example, the American Cancer Society has developed a guideline which recommends MRI as an adjunct to mammography screening for women at high risk, as identified by a risk prediction model [12]. 

Since the late 2000s, genome-wide association studies (GWAS) have identified hundreds of polymorphic loci associated with sporadic breast cancer risk [13,14,15,16,17,18,19,20,21,22,23,24,25,26,27,28,29]. Based on these results, various genetic risk models to assess personalized breast cancer risk have been developed [30]. Although genetic risk modelling by aggregation of the effects of multiple risk loci is a promising approach to stratifying individual risk, genetic risk assessment has yet to be used in modifying breast cancer prevention approaches [31]. 

We and others have speculated that feedback on genetic and environmental risk to individuals at high risk might be meaningful for breast cancer prevention [32,33]. In particular, genetic risk assessment in combination with environmental risk assessment might predict breast cancer risk better than either assessment alone. To date, however, only a few studies have attempted to predict individual breast cancer risk using both environmental factors and genetic factors [34,35,36]. 

Here, we aimed to develop a genetic risk score and integrate it with established risk factors for personalized risk assessment for breast cancer in Japanese.

## 2. Materials and Methods

### 2.1. Subjects

Breast cancer cases and corresponding controls from three hospital-based case–control studies were included in the study. The Nagano study was a multicenter, hospital-based case–control study of breast cancer conducted from May 2001 to September 2005 at four hospitals in Nagano Prefecture. Details of the study have been described previously [37]. Briefly, the case subjects were a consecutive series of women aged 20–74 years with newly diagnosed, histologically confirmed invasive breast cancer who were admitted to one of the four hospitals during the survey period. Of 412 eligible patients, 405 (98%) agreed to participate. Healthy controls were selected from medical checkup examinees in two of the hospitals and confirmed not to have any cancer. One control was matched with each case by age (within three years) and residential area (city or regional area) during the study period. Among potential control subjects, one declined to participate. Consequently, written informed consent was obtained from 405 matched pairs. Thereafter, two subjects refused to provide blood samples, and two declined use of their data outside the Nagano study. Due to a shortage of DNA samples, 12 pairs were excluded from the study. The Kagoshima study was a hospital-based case–control study conducted in two hospitals in Kagoshima City from May 2010 to March 2012. Cases were female patients with newly diagnosed and histologically confirmed breast cancer while controls were outpatients undergoing breast cancer screening who were confirmed without malignant disease. Consecutive cases admitted to either hospital during the study period were asked to participate in this study, and the participation rate was 91%. In total, 233 BC cases and 331 controls were analyzed, with written informed consent obtained from all. The Aichi study was conducted between 2001 and 2005 at Aichi Cancer Center Hospital [38,39]. This study was conducted within the framework of the Hospital-based Epidemiological Research Program in Aichi Cancer Center (HERPACC2). Cases were first-visit outpatients with histologically confirmed breast cancer during the study period. Controls were first-visit outpatients during the same period who were confirmed to have no malignancy and no history of neoplasia. Controls were selected randomly and matched by age at a case–control ratio of 1:2. All study subjects provided blood samples. Lifestyle factors were collected by self-administered questionnaire. 

In total, the present study included 1319 breast cancer cases and 2094 non-cancer controls. 

### 2.2. Evaluation of Environmental Risk Factors

Information on known environmental risk factors for breast cancer was collected by self-administered questionnaire in each study. Data from three studies were harmonized according to common items and a categorization of variables was defined. The following variables were considered as environmental risk factors: age at enrollment, body mass index (BMI, <18.5, 18.5–24.9, ≥25), ethanol drinking (never, <23 g/day, ≥23 g/day), cigarette smoking (never, ever), physical activity (yes, no), family history of breast cancer (yes, no), age at menarche (≤12 years old, 13 or 14 years old, ≥15 years old), parity (yes, no), number of children (0, 1–2, 3 or more), age at first birth (<30 years old, ≥30 years old, nonparous), breastfeeding (yes, no), hormone therapy (yes, no) and menopausal status (menstruation, menopause). BMI was calculated as the reported weight in kilograms divided by the reported height in meters squared. Ethanol consumption was estimated using the average number of alcohol beverages per day. Subjects reporting regular leisure time exercise at least once per month were classified as having physical activity. Family history was considered positive if a mother or a sister had ever had breast cancer. 

### 2.3. Statistical Analysis

We previously identified 23 breast cancer-associated SNPs reported in previous GWAS or candidate-gene association studies [17,18,19,20,21,22,23,24,25,26,27,28,29,40]. Additionally, 91 SNPs associated with breast cancer were identified in five GWAS studies [13,14,16,41]. The 114 identified loci are listed in Appendix A. 

Genomic DNA was extracted from the peripheral blood using a Qiagen FlexiGene DNA Kit (Qiagen, Hilden, Germany) in the Nagano study, a QIAamp DNA Blood Maxi Kit (Qiagen) in the Kagoshima study and a DNA Blood mini kit (Qiagen, Tokyo, Japan) in the Aichi study. The 114 loci were genotyped in the study subjects using SNPtype assays (Fluidigm, San Francisco, CA, USA). Among 114 loci, 11 monomorphic SNPs were excluded. Sixteen SNPs that were not accordant with the Hardy Weinberg Equilibrium (HWE) in at least one of the three populations were excluded. In total, 87 SNPs were included in further analysis. The impact of each SNP on breast cancer risk was evaluated by per allele odds ratio (OR) and 95% confidence interval (CI) using a logistic regression model adjusted for age at enrollment. The results of the three studies were combined using random effects meta-analysis. SNPs with summary *p*-values less than 0.05 were selected for risk prediction modeling. Linkage disequilibriums (LD) of SNPs located within same genes were calculated. LD of the candidate loci clustered in the same region were assessed by Haploview 4.2 [42]. Strong LD was defined as a one-sided upper 95% confidence bound on D’ of more than 0.98 and a lower 95% confidence bound above 0.7. SNPs within the same LD block were excluded, except one SNP with the lowest *p*-value for breast cancer risk. Similarly, SNPs with summary *p*-values less than 0.10 and 0.30 were also used in sensitivity analysis for genetic risk modeling. 

The genetic risk group for breast cancer was defined according to the number of risk alleles in each control subject. Three risk groups (Low, Moderate, High) were defined by the distribution of risk allele numbers. Approximately 20%, 70% and 10% of controls were defined as the low-, moderate- and high-risk group, respectively. Breast cancer susceptibility in each risk group was evaluated by OR and its 95% CI using both crude and adjusted logistic regression models. Age at enrollment was adjusted in the crude model. In addition to the crude model, environmental risk factors were included in the adjusted model. ORs in total populations were calculated by crude and adjusted models with the addition of study site as a covariate. To assess the discriminatory ability of the risk prediction model, the area under the curve (AUC) in the Receiver Operating Characteristic (ROC) curve—also known as the concordance statistic (C-statistic)—was used. The C-statistic in the genetic model for each study population and the total population was calculated using logistic regression models which included the genetic risk score in the risk model. Similarly, C-statistic in the environmental model was calculated using logistic regression models which included the environmental risk factors. All variables in the genetic and environmental models were included in the inclusive model. In the ROC, the y axis shows sensitivity and the x axis shows the false positive rate, with AUC values ranging from 0.5 to 1. The straight line in the ROC shows a random classification of case and control subjects with an AUC of 0.5, while an AUC value of 1 corresponds to a perfect classification. An AUC value between 0.7 and 0.8 is acceptable while a value greater than 0.8 represents excellent model discrimination [43]. In addition to the genetic risk score model, we also assessed the genetic risk score model in three levels and the allelic risk model as sensitivity analyses. The C-statistic of the genetic risk score model in three levels was calculated using a logistic regression model which included the low, moderate and high genetic risk groups. The C-statistic of the allelic risk model was calculated using logistic regression models which included the summation of logarithmic allelic risk ORs of SNPs in the genetic risk models. The C-statistic values were compared using the method of DeLong et al. [44]. A calibration of the risk model was assessed by the Hosmer–Lemeshow goodness-of-fit statistic and calibration plots [45]. Subjects were grouped by decile of predicted probability. A significant *p*-value in the Hosmer–Lemeshow test indicates disagreement between the predicted and observed outcomes. The mean predicted probability was plotted against the mean observed probability for each decile in a calibration plot. A *p*-value < 0.05 was defined as the threshold of significance. Statistical analyses were conducted using Stata version 15.2 (StataCorp LP, College Station, TX, USA).

## 3. Results

The three case–control studies are characterized in Table 1. In total, 1319 cases and 2114 controls were included in the present study, broken down as 389 and 389 from the Nagano study, 233 and 331 from the Kagoshima study and 697 and 1394 from the Aichi study, respectively. Due to matching, age distributions among cases and controls in the Nagano and Aichi studies were not different, whereas cases in the Kagoshima study were older than controls. The proportion of obesity (BMI 25 or more) was similar in the Nagano and Aichi studies, but obesity was more prevalent in cases in the Kagoshima study. Finally, hormone therapy use was more prevalent in controls in the Kagoshima and Aichi studies. 

Among 114 genotyped breast cancer susceptibility loci identified by GWAS studies (Appendix A), 19 SNPs had statistically significant summary *p*-values of less than 0.05. Five loci located in 10q26 (rs2981579, rs2981578, rs1219648, rs2420946 and rs2981582) and two in 16q12 (rs3803662 and rs4784227) were in strong LD. Four loci in 10q26 (rs2981578, rs1219648, rs2420946 and rs2981582) and one in 16q12 (rs3803662) were excluded from further analysis. The list of breast cancer susceptibility loci and their allelic ORs is shown in Table 2. Similarly, 22 SNPs with summary *p*-values of less than 0.10 and 42 SNPs with summary *p*-values of less than 0.30 were selected for additional genetic risk assessment. 

Genetic risk groups were defined according to the risk allele distribution of the 14 SNPs in controls (Figure 1), with those with 0 to 10, 11 to 16 and 17 to 28 risk alleles defined as low-, moderate- and high-risk groups, respectively. Subjects with undetermined alleles were classified as undetermined. Subject proportions in the low-, moderate- and high-risk groups were 23.84%, 69.30% and 6.86%, respectively. Proportions in risk groups in each study’s controls were similar to those in the total control subjects. In the crude model, summary ORs of breast cancer in the moderate- and high-risk groups were 1.70 (95% CI, 1.41–2.05) and 3.29 (CI, 2.49–4.34) compared with low-risk group, respectively. The ORs in each study were similar to those in the total population. ORs were similar after adjustment for known breast cancer risk factors.

Figure 2 shows the ROC curves of the genetic, environmental and inclusive risk models in the three study populations and total population. The C-statistics of genetic model, environmental model and inclusive models in the three populations and total population are shown in Table 3. The C-statistics of the genetic models were 0.605, 0.609, 0.604 and 0.633 in the Nagano, Kagoshima, Aichi and overall populations, respectively. The C-statistics of the inclusive model (combination of genetic and environmental models) in the Nagano, Kagoshima, Aichi and total populations were better than those of the environmental models. The ROC curves in total population resembled those in the Aichi study, because of the relatively large sample size of the Aichi study. A calibration plot of the inclusive model in the overall population remained close to the ideal calibration line (calibration slope of 1.02 and *p* for Hosmer–Lemeshow test = 0.506) (Appendix A). 

The impact of risk groups stratified by menopausal status is shown in Appendix A. The ORs among premenopausal females in the moderate- and high-risk groups were 2.03 (CI, 1.52–2.72) and 4.27 (CI, 2.78–6.56), respectively. For postmenopausal females, the ORs in the moderate- and high-risk groups were 1.46 (CI, 1.13–1.90) and 2.69 (CI, 1.83–3.95), respectively. The C-statistics of the genetic model in premenopausal and postmenopausal females were 0.652 (CI, 0625–0.680) and 0.621 (CI, 0.594–0.647), respectively (Appendix A). The C-statistics were significantly improved with the combination of genetic and environmental models in both premenopausal and postmenopausal females.

To check the validity of the SNP selection in the genetic risk model, genetic risk models with additional SNPs were assessed (Appendix A). The C-statistics of genetic risk models that included 14, 22 and 42 SNPs in the total population were 0.633 (95% CI 0.614–0.652), 0.636 (95% CI 0.617–0.655) and 0.636 (95% CI 0.617–0.655), respectively. Accordingly, the C-statistic of the genetic risk model with 14 SNPs was not statistically poorer than that of those with 22 or 42 SNPs, and the inclusion of more SNPs did not improve model performance. 

To assess the validity of genetic risk categorization, C-statistics of the three levels of the genetic risk model, genetic risk score model (number of risk alleles) and allelic risk model (summation of logarithmic allelic ORs) were assessed (Appendix A). The C-statistics of the genetic risk score models did not significantly differ from those of the allelic risk models.

## 4. Discussion

We established a genetic risk model for breast cancer in subjects from three case–control studies in Japan using 14 risk loci identified in GWASs. The high-risk group, which accounted for 6.86% of total population, had a 3.27 times higher breast cancer risk than the low-risk group. While the discriminatory ability of the genetic risk model alone was not satisfactory, its combination with an environmental risk model produced significantly improved performance. Further, performance of the combined risk model was consistent between premenopausal and postmenopausal females. 

Many GWASs aimed at breast cancer risk seek to elucidate carcinogenic mechanisms. However, these studies have no direct impact on clinical practice [46]. One reason is the small impact of each loci. In our study also, the magnitude of each SNP on the risk of breast cancer was small. When aggregated, however, these risk alleles together would likely indicate substantial risk elevation in those in the high-risk group. In previous studies in Japanese females, the impact of genetic risk in the high-risk groups was larger than that of cigarette smoking, alcohol drinking and obesity [47,48,49,50]. Unlike smoking, drinking and obesity, however, genetic risk cannot be modified. Nevertheless, preventive approaches for women with high genetic risk should be considered. 

The use of genetic risk stratification for breast cancer prevention should be investigated. Several studies have assessed preventive approaches to genetic risk [51,52]. One possible strategy is personalized breast cancer screening: currently, biannual mammography is recommended for Japanese women aged 40 years or older [53], but screening intensity might be strengthened in high-risk individuals. Appropriate frequency, examination modalities, and age of screening initiation by predicted individual breast cancer risk should be investigated. A second potential strategy is lifestyle modification via individual risk feedback. Feedback on genetic risk in combination with education about a healthy lifestyle might induce individuals to modify behaviors associated with breast cancer risk such as obesity, physical activity, alcohol drinking, and cigarette smoking. Appropriate lifestyle modification directly decreases breast cancer risk [54], while a healthy lifestyle attenuates genetic breast cancer risk [55]. Lifestyle modification is difficult to achieve and sustain, and few studies have reported success in using risk feedback to change lifestyle. In addition, the impact of risk feedback and that of intervention for each modifiable risk factor must almost certainly differ. Thus, weight management, physical activity, abstinence from drinking, and smoking cessation should be recommended with appropriate intervention strategies. Novel and personalized risk communications and interventions suitable for lifestyle modification should be investigated. 

Compared to the allelic risk model, the risk score model appears to have had an attenuated discriminatory ability. The risk score model was based on the assumption that all risk alleles confer the same magnitude of breast cancer risk. While the allelic ORs of risk loci ranged from 1.11 to 1.46 in the total population, the C-statistics of risk score models were not poorer than those of allelic risk models, indicating that the risk score models could be used in place of the allelic risk models. The number of risk alleles and the three corresponding risk grouping levels are simple to implement and comprehensive for females in general populations, the characteristics which would facilitate preventive interventions for breast cancer. The risk model would be available in other populations, although useful sets of alleles must be assessed in the populations. Randomized controlled studies to determine whether genetic risk feedback modifies individual behavior for breast cancer prevention are warranted.

Hundreds of loci associated with breast cancer risk have been identified in GWAS studies. A previous polygenic risk model based on a large GWAS dataset reported AUCs of 0.603, 0.630 and 0.636 using 77, 313 and 3820 SNPs, respectively [56]. These findings suggested that using more SNPs associated with breast cancer risk might improve model performance; in our present study, however, the inclusion of SNPs with low significance did not improve performance: genetic models with 22 and 44 SNPs offered no significant improvement over that with 14 SNPs. Indeed, three GWAS studies in Japanese populations identified only 31 loci in 19 regions [57,58,59]. Attempts to further improve the performance of genetic models by adding more SNPs would, therefore, require a larger sample size. 

The major strength of this study was its study population. Because we established the risk model using three case–control studies conducted in geographically distant regions, the results are generalizable to the Japanese population. Nevertheless, the study was based on hospital-based case–control studies, meaning that several methodological limitations exist. First, the values for self-reported lifestyle factors considered as potential confounding factors might have some misclassification and recall bias. Second, selection bias in study subjects is inevitable in hospital-based case–control studies, and external validity should be interpreted carefully. Distributions of alcohol drinking and cigarette smoking in controls were highly consistent with those in a national survey [60], suggesting that our study population did not vary from the Japanese general population in these regards. Third, we were unable to establish a risk prediction model by tumor subtype as not all cases had information on hormonal status. Against this, however, establishment of a risk model according to tumor subtype would have little meaning in personalized risk assessment for breast cancer. Fourth, categorization of high genetic risk had no consensus. Further studies were required to evaluate reasonable and useful thresholds of genetically high-risk groups.

## 5. Conclusions

This genetic risk score model using 14 GWAS-identified loci in combination with environmental factors is able to stratify breast cancer risk. New breast cancer prevention strategies for genetically high-risk populations should be developed. 

## Figures and Tables

**Figure 1 cancers-13-03796-f001:**
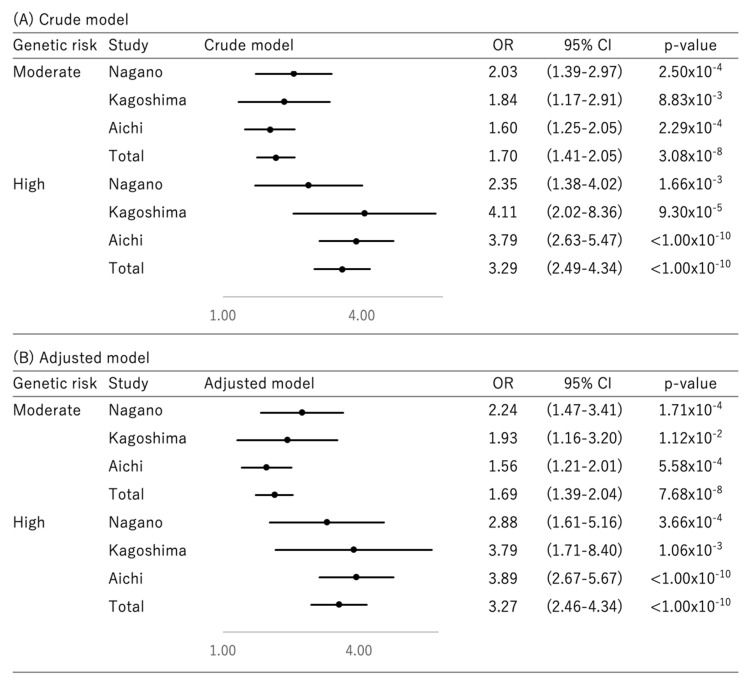
Odds ratio of breast cancer risk in each risk group; OR; odds ratio, CI; confidence interval. (**A**) Age was included in the crude model. (**B**) Age, BMI, ethanol intake, smoking, physical activity, family history of breast cancer, age at menarche, parity, number of births, age at first birth, breastfeeding and hormone therapy were included in the adjusted model. Genetic risk group was defined by the number of risk alleles, with 0–10, 11–16 and 17–28 risk alleles defined as the low-, moderate- and high-risk groups, respectively.

**Figure 2 cancers-13-03796-f002:**
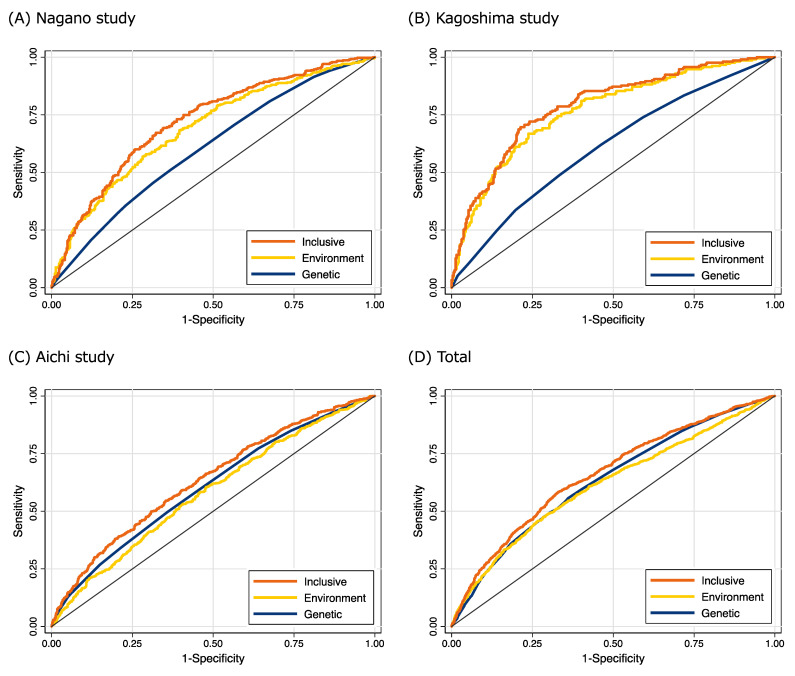
ROC curves of genetic, environmental and inclusive risk models in (**A**) Nagano study, (**B**) Kagoshima study, (**C**) Aichi study, and (**D**) total population. Orange, yellow and navy lines are ROC curves of the Inclusive, Environmental and Genetic models, respectively. Age, body mass index, ethanol drinking, cigarette smoking, physical activity, family history of breast cancer, age at menarche, parity, number of children, age at first delivery, breastfeeding, hormone therapy and menopausal status were adjusted in the Environmental model.

**Table 1 cancers-13-03796-t001:** Characteristics of study participants.

		Cases								Controls							
		Nagano (%)		Kaghoshima (%)		Aichi (%)		Total (%)		Nagano (%)		Kaghoshima (%)		Aichi (%)		Total (%)	
Age																	
	−39	33	(8.48)	13	(5.58)	105	(15.06)	151	(11.45)	26	(6.68)	79	(23.87)	204	(14.63)	309	(14.62)
	40–49	112	(28.79)	43	(18.45)	197	(28.26)	352	(26.69)	107	(27.51)	96	(29.00)	416	(29.84)	619	(29.28)
	50–59	122	(31.36)	56	(24.03)	222	(31.85)	400	(30.33)	135	(34.70)	86	(25.98)	429	(30.77)	650	(30.75)
	60–69	88	(22.62)	55	(23.61)	132	(18.94)	275	(20.85)	96	(24.68)	41	(12.39)	271	(19.44)	408	(19.30)
	70+	34	(8.74)	62	(26.61)	41	(5.88)	137	(10.39)	25	(6.43)	29	(8.76)	74	(5.31)	128	(6.05)
	UK	0	(0.00)	4	(1.72)	0	(0.00)	4	(0.30)	0	(0.00)	0	(0.00)	0	(0.00)	0	(0.00)
BMI																	
	<18.5	30	(7.71)	11	(4.72)	56	(8.03)	97	(7.35)	13	(3.34)	32	(9.67)	116	(8.32)	161	(7.62)
	18.5–24.9	275	(70.69)	145	(62.23)	498	(71.45)	918	(69.60)	284	(73.01)	249	(75.23)	1021	(73.24)	1554	(73.51)
	≥25	84	(21.59)	66	(28.33)	143	(20.52)	293	(22.21)	92	(23.65)	50	(15.11)	245	(17.58)	387	(18.31)
	UK	0	(0.00)	11	(4.72)	0	(0.00)	11	(0.83)	0	(0.00)	0	(0.00)	12	(0.86)	12	(0.57)
Ethanol intake																	
	Never	288	(74.04)	182	(78.11)	509	(73.03)	979	(74.22)	272	(69.92)	248	(74.92)	1027	(73.67)	1547	(73.18)
	<23 g/day	72	(18.51)	37	(15.88)	148	(21.23)	257	(19.48)	88	(22.62)	68	(20.54)	299	(21.45)	455	(21.52)
	≥23 g/day	27	(6.94)	9	(3.86)	33	(4.73)	69	(5.23)	29	(7.46)	14	(4.23)	51	(3.66)	94	(4.45)
	UK	2	(0.51)	5	(2.15)	7	(1.00)	14	(1.06)	0	(0.00)	1	(0.30)	17	(1.22)	18	(0.85)
Smoking																	
	Never	307	(78.92)	183	(78.54)	585	(83.93)	1075	(81.50)	358	(92.03)	283	(85.50)	1113	(79.84)	1754	(82.97)
	Ever	78	(20.05)	46	(19.74)	110	(15.78)	234	(17.74)	30	(7.71)	48	(14.50)	278	(19.94)	356	(16.84)
	UK	4	(1.03)	4	(1.72)	2	(0.29)	10	(0.76)	1	(0.26)	0	(0.00)	3	(0.22)	4	(0.19)
Physical activity																	
	No	337	(86.63)	128	(54.94)	416	(59.68)	881	(66.79)	324	(83.29)	176	(53.17)	862	(61.84)	1362	(64.43)
	Yes	47	(12.08)	95	(40.77)	281	(40.32)	423	(32.07)	64	(16.45)	153	(46.22)	532	(38.16)	749	(35.43)
	UK	5	(1.29)	10	(4.29)	0	(0.00)	5	(0.38)	1	(0.26)	2	(0.60)	0	(0.00)	3	(0.14)
Family history of breast cancer																	
	No	323	(83.03)	195	(83.69)	632	(90.67)	1150	(87.19)	350	(89.97)	296	(89.43)	1305	(93.62)	1951	(92.29)
	Yes	40	(10.28)	30	(12.88)	65	(9.33)	135	(10.24)	25	(6.43)	30	(9.06)	89	(6.38)	144	(6.81)
	UK	26	(6.68)	8	(3.43)	0	(0.00)	34	(2.58)	14	(3.60)	5	(1.51)	0	(0.00)	19	(0.90)
Age at menarche																	
	≤12 y.o.	139	(35.73)	63	(27.04)	216	(30.99)	418	(31.69)	145	(37.28)	115	(34.74)	439	(31.49)	699	(33.07)
	13–14 y.o.	164	(42.16)	106	(45.49)	340	(48.78)	610	(46.25)	175	(44.99)	155	(46.83)	648	(46.48)	978	(46.26)
	≥15 y.o.	85	(21.85)	57	(24.46)	133	(19.08)	275	(20.85)	69	(17.74)	59	(17.82)	277	(19.87)	405	(19.16)
	UK	1	(0.26)	7	(3.00)	8	(1.15)	16	(1.21)	0	(0.00)	2	(0.60)	30	(2.15)	32	(1.51)
Parity																	
	No	15	(3.86)	32	(13.73)	107	(15.35)	154	(11.68)	4	(1.03)	69	(20.85)	203	(14.56)	276	(13.06)
	Yes	334	(85.86)	175	(75.11)	589	(84.51)	1098	(83.24)	336	(86.38)	249	(75.23)	1188	(85.22)	1773	(83.87)
	UK	40	(10.28)	26	(11.16)	1	(0.14)	67	(5.08)	49	(12.60)	13	(3.93)	3	(0.22)	65	(3.07)
Number of births																	
	nonparous	15	(3.86)	32	(13.73)	107	(15.35)	154	(11.68)	4	(1.03)	69	(20.85)	203	(14.56)	276	(13.06)
	1 or 2	237	(60.93)	115	(49.36)	452	(64.85)	804	(60.96)	213	(54.76)	155	(46.83)	867	(62.20)	1235	(58.42)
	≥3	97	(24.94)	60	(25.75)	137	(19.66)	294	(22.29)	123	(31.62)	94	(28.40)	315	(22.60)	532	(25.17)
	UK	40	(10.28)	26	(11.16)	1	(0.14)	67	(5.08)	49	(12.60)	13	(3.93)	9	(0.65)	71	(3.36)
Age at first birth																	
	<30 y.o.	137	(35.22)	78	(33.48)	270	(38.74)	485	(36.77)	148	(38.05)	89	(26.89)	653	(46.84)	890	(42.10)
	≥30 y.o.	197	(50.64)	95	(40.77)	316	(45.34)	608	(46.10)	188	(48.33)	160	(48.34)	522	(37.45)	870	(41.15)
	nonparous	15	(3.86)	32	(13.73)	108	(15.49)	155	(11.75)	4	(1.03)	69	(20.85)	207	(14.85)	280	(13.25)
	UK	40	(10.28)	28	(12.02)	3	(0.43)	71	(5.38)	49	(12.60)	13	(3.93)	12	(0.86)	74	(3.50)
Breastfeeding																	
	No	72	(18.51)	17	(7.30)	131	(18.79)	220	(16.68)	67	(17.22)	21	(6.34)	260	(18.65)	348	(16.46)
	Yes	306	(78.66)	158	(67.81)	558	(80.06)	1022	(77.48)	322	(82.78)	229	(69.18)	1117	(80.13)	1668	(78.90)
	UK	11	(2.83)	58	(24.89)	8	(1.15)	77	(5.84)	0	(0.00)	81	(24.47)	17	(1.22)	98	(4.64)
Hormone therapy																	
	No	327	(84.06)	203	(87.12)	603	(86.51)	1133	(85.90)	331	(85.09)	265	(80.06)	1141	(81.85)	1737	(82.17)
	Yes	55	(14.14)	22	(9.44)	88	(12.63)	165	(12.51)	55	(14.14)	63	(19.03)	229	(16.43)	347	(16.41)
	UK	7	(1.80)	8	(3.43)	6	(0.86)	21	(1.59)	3	(0.77)	3	(0.91)	24	(1.72)	30	(1.42)

BMI; body mass index, UK; unknown, y.o.; years old.

**Table 2 cancers-13-03796-t002:** SNPs with a significant association with breast cancer.

			Total			Nagano Study			Kagoshima Study			Aichi Study		
SNP	Chromosome	Risk/Reference Allele	OR	95% CI	*p*	OR	95% CI	*p*	OR	95% CI	*p*	OR	95% CI	*p*
rs4849887	2q14.2	C/T	1.16	(1.00–1.34)	0.046	0.97	(0.73–1.28)	0.820	1.38	(0.93–2.06)	0.111	1.21	(1.00–1.46)	0.047
rs10931936	2q33.1	T/C	1.11	(1.00–1.23)	0.047	1.15	(0.93–1.40)	0.192	1.20	(0.91–1.58)	0.200	1.08	(0.94–1.23)	0.278
rs16857609	2q35	T/C	1.14	(1.03–1.26)	0.012	0.97	(0.80–1.19)	0.803	1.29	(1.00–1.67)	0.052	1.18	(1.03–1.35)	0.015
rs4973768	3p24.1	T/C	1.15	(1.02–1.30)	0.028	0.93	(0.73–1.18)	0.552	1.40	(1.03–1.90)	0.034	1.20	(1.02–1.42)	0.028
rs7697216	4q34.1	C/T	1.26	(1.10–1.44)	0.001	1.30	(0.99–1.71)	0.056	1.26	(0.91–1.74)	0.171	1.24	(1.04–1.48)	0.015
rs1432679	5q33.3	C/T	1.13	(1.01–1.25)	0.028	1.15	(0.93–1.43)	0.197	1.14	(0.87–1.50)	0.338	1.11	(0.97–1.28)	0.127
rs2046210	6q25.1	T/C	1.23	(1.11–1.37)	1.53 × 10^−4^	1.09	(0.88–1.35)	0.441	1.20	(0.90–1.60)	0.203	1.30	(1.13–1.49)	1.84 × 10^−4^
rs13365225	8p11.23	A/G	1.11	(1.00–1.23)	0.043	1.15	(0.93–1.41)	0.188	1.23	(0.95–1.59)	0.124	1.07	(0.94–1.22)	0.324
rs13281615	8q24.21	G/A	1.13	(1.02–1.25)	0.022	1.00	(0.81–1.22)	0.984	1.15	(0.88–1.49)	0.307	1.18	(1.04–1.35)	0.014
rs2981579	10q26.13	G/A	1.22	(1.10–1.34)	1.42 × 10^−4^	1.37	(1.13–1.67)	0.002	1.21	(0.93–1.56)	0.152	1.15	(1.01–1.32)	0.031
rs17271951	16q12.1	C/T	1.41	(1.24–1.59)	4.89 × 10^−8^	1.34	(1.06–1.70)	0.016	1.58	(1.14–2.19)	0.006	1.40	(1.19–1.64)	3.40 × 10^−5^
rs4784227	16q12.1	T/C	1.46	(1.30–1.64)	2.47 × 10^−10^	1.52	(1.20–1.92)	4.48 × 10^−4^	1.63	(1.19–2.23)	0.002	1.40	(1.20–1.62)	1.07 × 10^−5^
rs8051542	16q12.1	T/C	1.21	(1.07–1.36)	0.002	1.08	(0.85–1.37)	0.533	1.39	(1.01–1.92)	0.043	1.23	(1.05–1.43)	0.010
rs11075995	16q12.2	A/T	1.18	(1.06–1.31)	0.003	1.25	(1.01–1.56)	0.041	1.11	(0.85–1.44)	0.461	1.16	(1.02–1.33)	0.028

SNP; single nucleotide polymorphism, OR; odds ratio, CI; confidence interval.

**Table 3 cancers-13-03796-t003:** C-statistics of genetic, environmental and inclusive risk models.

	Genetic		Environment		Inclusive		
	C-Statistics	95% CI	C-Statistics	95% CI	C-Statistics	95% CI	*p*-Value
Nagano	0.605	(0.566–0.645)	0.691	(0.654–0.728)	0.721	(0.685–0.757)	0.005
Kagoshima	0.609	(0.560–0.657)	0.767	(0.726–0.808)	0.789	(0.750–0.828)	0.018
Aichi	0.604	(0.579–0.630)	0.581	(0.555–0.607)	0.635	(0.610–0.660)	7.05 × 10^−6^
Total	0.633	(0.614–0.652)	0.616	(0.596–0.636)	0.659	(0.640–0.678)	1.67 × 10^−9^

Genetic risk group was included in the Genetic model. Age, BMI, ethanol intake, smoking, physical activity, family history of breast cancer, age at menarche, parity, number of births, age at first birth, breastfeeding and hormone therapy were included in the Environmental model. All variables in the Genetic model and the Environmental model were included in the Inclusive model. *p*-values were calculated by testing differences between the Environmental model and the Inclusive model.

## Data Availability

The data are not publicly available due to ethical and data security requirements.

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
