# Peer review of "A Personal Breast Cancer Risk Stratification Model Using Common Variants and Environmental Risk Factors in Japanese Females"

_cancers, 2021, doi:10.3390/cancers13153796_

Round 1

Reviewer 1 Report

Summary

The authors set out to construct a predictive model for breast cancer that combines both genetic and environmental factors to enhance individualized prevention strategies. Three case-control studies conducted in Japan involving genetic tests and lifestyle questionnaires were combined in the present analysis. Thirteen environmental risk factors were analyzed along with 14 genetic loci (SNPs) that were deemed the most predictive of disease. Strengths of this study include the relatively large number of participants, the multi-site collection of data, and the large number of environmental factors evaluated.

I think the authors make a strong argument that evaluation of both genetic and environmental (lifestyle) factors together can provide a more predictive model than either set of factors alone. The study also highlights the fact that genetic predispositions can be attenuated by positive lifestyle choices, and that this data may be useful in genetic counseling for patients at high risk of developing breast cancer.

Minor recommended changes:

Figure 1: A dotted line is mentioned in the legend, but I do not see a dotted line in the figures.

Results (Discussion of Figure 1): In the Aichi and Total graphs, the genetic and environment curves are very similar, whereas in the Nagano and Kagoshima graphs, these curves are quite different. What accounts for the differences between studies?

Author Response

We appreciate the reviewer for the valuable and thoughtful comments. We revised our manuscript according to the comments. Each comment was helpful to improve the article.

Reply to Reviewer 1

Comment 1. Figure 1: A dotted line is mentioned in the legend, but I do not see a dotted line in the figures.

Response:

I was sorry for the wrong figure legend. The legend in Figure 1 was replaced with that in supplemental Figure 1. We revised the legend as follows; “Orange, yellow and navy lines are ROC curves of the Inclusive, Environmental and Genetic models, respectively. Age, body mass index, ethanol drinking, cigarette smoking, physical activity, family history of breast cancer, age at menarche, parity, number of children, age at first delivery, breastfeeding, hormone therapy and menopausal status were adjusted in the Environmental model.”

Comment 2. Results (Discussion of Figure 1): In the Aichi and Total graphs, the genetic and environment curves are very similar, whereas in the Nagano and Kagoshima graphs, these curves are quite different. What accounts for the differences between studies?

Response:

The number of subjects in each study would affect the graph of AUC in overall population. Nagano, Kagoshima and Aichi study included 778, 564, and 2091 subjects. Thus, approximately 60% of subjects were Aichi study participants. We added the sentence below to explain it; "the ROC curves in total population resembled those in Aichi study, because of the relatively large sample size of Aichi study."

Reviewer 2 Report

The manuscript by Oze et al. presents analyses for breast cancer risk assessment based on GWAS identified common variants and combining with environmental risk factors. Overall, the results presented appear substantial in scope for breast cancer risk assessment for people of Japanese ancestry. The sample size from the three studies is appropriate. Authors need to address the following comments for improving the manuscript:

  1. It is unclear why the authors chose three risk groups. Providing a basis for this grouping is required for future replication of the study.
  2. For cancer risk assessments based on common variants, specifically involving polygenic risk scores (PRS), it is essential to perform heritability analysis. It is recommended to perform PRS analyses using GWAS data with an h2SNP > 0.05. If  h2SNP estimates were not made before, use tools like LD Score regression or SumHer for such evaluation. 
  3. In the tables provided, specifically in Supplementary Table 1, authors should include information about effect allele. If the allelic direction is misrepresented in PRS calculations, it can lead to spurious results. 
  4. Graphical representation is inadequate. Please refer to Chio et al., paper (PMID: 32709988) for guidelines on presenting disease risk assessment data in a graphical format.

Author Response

We appreciate the reviewer for your valuable and thoughtful comments. We revised our manuscript according to the comments. Each comment was helpful to improve the article.

Reply to Reviewer 2

Comment 1. It is unclear why the authors chose three risk groups. Providing a basis for this grouping is required for future replication of the study.

Response:

We appreciate your important comment. There was no consensus of a definition of “high genetic risk”. We defined the controls with middle 70% of the risk score as moderate risk group. Remaining higher 10% and lower 20% were defined as high and low risk groups, because deciles were often used for PRS categorization. As the reviewer pointed out, further studies are required to find reasonable and useful grouping. Therefore, we added the sentences as limitations; "categorization of high genetic risk had no consensus. Further studies were required to evaluate reasonable and useful threshold of genetically high risk groups."

Comment 2. For cancer risk assessments based on common variants, specifically involving polygenic risk scores (PRS), it is essential to perform heritability analysis. It is recommended to perform PRS analyses using GWAS data with an h2SNP > 0.05. If h2SNP estimates were not made before, use tools like LD Score regression or SumHer for such evaluation.

Response:

We appreciate your suggestion. We agreed that heritability was an important measure for PRS. Because the genotyping of 114 loci, not the genome-wide scanning, was performed in this study, we were not able to perform PRS to evaluate SNP heritability. Ishigaki K et al. reported the breast cancer heritability in Japanese population was 0.04 (Ishigaki K et al. Nat Genet 2020 PMID: 32514122). It might suggest the heritability in Japanese population was around 0.04.

Comment 3. In the tables provided, specifically in Supplementary Table 1, authors should include information about effect allele. If the allelic direction is misrepresented in PRS calculations, it can lead to spurious results.

Response:

We appreciate the advice. According to the reviewer’s comment, we added a column of reference and risk alleles in supplemental table 1.

Comment 4. Graphical representation is inadequate. Please refer to Choi et al., paper (PMID: 32709988) for guidelines on presenting disease risk assessment data in a graphical format.

Response:

We appreciate your suggestion. The article recommended that the results of PRS association tests are displayed using bar and quantile plots. We revised table 3 using bar and plots by genetic risk groups. (Figure 1 in the revised manuscript).

Reviewer 3 Report

Identification of high breast cancer risk groups, allowing for personalised approach to breast cancer  prevention, is still the unmet medical need and seems to stay a bit behind compared to also developing field of personalised cancer treatment. Therefore, development and validation of the methods allowing to predict high-risk individuals is highly appreciated. In their manuscript, the Authors describe the new risk model allowing to assess breast cancer risk arising both from the genetics, the lifestyle of an individual, and both. Reading the manuscript, the following questions arouse:

  1. If and to how extent, in the Authors’ opinion, an appropriate lifestyle (environmental factors) could lower the risk of breast cancer risk in the individuals with high risk of the disease because of their genetics (i.e. harbouring BRCA1 and 2 or other mutations).
  2. Is high genetic risk of breast cancer really modifiable?
  3. What particular preventive approaches (apart from mastectomy, and examination) the Authors would recommend to women with high breast cancer risk (i.e. lower BMI, do exercise, enrich their diet with antioxidants…).
  4. Do the Authors consider implementation and further validation of their model of personalised breast cancer risk assessment in the clinical practice? It could be possible to validate it further in clinic considering the fact that 14 different medical-scientific departments were involved in this study
  5. Do the Authors think that their model could be useful in determination of breast cancer risk in other than Japanese populations?

Author Response

We appreciate the reviewer for your valuable and thoughtful comments. We revised our manuscript according to the comments. Each comment was helpful to improve the article.

Reviewer 3

Comment 1. If and to how extent, in the Authors’ opinion, an appropriate lifestyle (environmental factors) could lower the risk of breast cancer risk in the individuals with high risk of the disease because of their genetics (i.e. harboring BRCA1 and 2 or other mutations).

Response:

The association between lifestyles and the familial breast cancer mutations like BRCA1/2 was crucial problem. Because evidence was limited, further studies were warranted. We evaluated common variants with small effects for breast cancer. The aggregation of variants with small effects could stratify sporadic breast cancer risk. Our study did not evaluate hereditary (familial) breast cancer, thus, rare and high risk germline mutations including BRCA1/2 were not assessed. Therefore, the study results could not suggest the question.

Comment 2. Is high genetic risk of breast cancer really modifiable?

Response:

We think genetic risk was difficult to modify. As we described in the 3rd paragraph of the discussion, healthy lifestyles might possibly attenuate the genetic risk. However, lifestyle modification was difficult to achieve and sustain. Therefore, personalized and intensive screening for high risk females were possible strategy.

Comment 3. What particular preventive approaches (apart from mastectomy, and examination) the Authors would recommend to women with high breast cancer risk (i.e. lower BMI, do exercise, enrich their diet with antioxidants…).

Response:

As we described in discussion (line 300-302), “feedback on genetic risk in combination with education about a healthy lifestyle might induce individuals to modify behaviors associated with breast cancer risk such as obesity, physical activity, alcohol drinking, and cigarette smoking.” Therefore, weight management, exercise, quit alcohol, and smoking cessation were recommendation for high risk females. To explain it more clearly, we added the sentence “weight management, physical activity, abstinence from drinking, and smoking cessation should be recommended with appropriate intervention strategies.”

Comment 4. Do the Authors consider implementation and further validation of their model of personalized breast cancer risk assessment in the clinical practice? It could be possible to validate it further in clinic considering the fact that 14 different medical-scientific departments were involved in this study

Response:

We appreciate your suggestion. We described in the discussion as follows; “Randomized controlled studies to determine whether genetic risk feedback modifies individual behavior for breast cancer prevention are warranted.” We are conducting the randomized controlled study whether genetic risk feedback modifies individual cancer prevention behavior (UMIN000026768) as a next step.

Comment 5. Do the Authors think that their model could be useful in determination of breast cancer risk in other than Japanese populations?

Response:

We appreciate your suggestion. We think the genetic risk model would be useful to stratify breast cancer risk in other populations. However, alleles associated with breast cancer risk would vary between populations. Therefore, it is required to assess the useful set of alleles for the genetic risk stratification in each population. We added the sentence; “the risk model would be available in other populations, although useful set of alleles must be assessed in the populations.”